# Prolonged Dystocic Labor in Neuraxial Analgesia and the Role of Enkephalin Neurotransmitters: An Experimental Study

**DOI:** 10.3390/ijms24043767

**Published:** 2023-02-13

**Authors:** Antonio Malvasi, Ettore Cicinelli, Giorgio Maria Baldini, Antonella Vimercati, Renata Beck, Miriam Dellino, Gianluca Raffaello Damiani, Gerardo Cazzato, Eliano Cascardi, Andrea Tinelli

**Affiliations:** 1Department of Biomedical and Human Oncological Science (DIMO), Unit of Obstetrics and Gynecology, University of Bari, 70121 Bari, Italy; 2Momò Fertilfe Clinic, 76011 Bisceglie, Italy; 3Department of Medical and Surgical Sciences, Anesthesia and Intensive Care Unit, Policlinico Riuniti Foggia, University of Foggia, 71122 Foggia, Italy; 4Section of Pathology, Department of Emergency and Organ Transplantation (DETO), University of Bari “AldoMoro”, 70124 Bari, Italy; 5Department of Medical Sciences, University of Turin, 10124 Turin, Italy; 6Pathology Unit, FPO-IRCCS, Candiolo Cancer Institute, 10060 Candiolo, Italy; 7Department of Obstetrics and Gynecology, CERICSAL (CEntro di RIcerca Clinico SALentino), Veris delli Ponti Hospital, 73020 Scorrano, Italy

**Keywords:** enkephalinergic neuro fibers, dystocia, prolonged labor, delivery, cesarean section, asynclitism

## Abstract

The investigation studied the enkephalinergic neuro fibers (En) contained in the Lower Uterine Segment (LUS) during the prolonged dystocic labor (PDL) with Labor Neuraxial Analgesia (LNA). PDL is generally caused by fetal head malpositions in the Occiput Posterior Position (OPP), Persistent Occiput Posterior Position (POPP), in a transverse position (OTP), and asynclitism (A), and it is detected by Intrapartum Ultrasonography (IU). The En were detected in the LUS samples picked up during cesarean section (CS) of 38 patients undergoing urgent CS in PDL, compared to 37 patients submitted to elective CS. Results were statistically evaluated to understand the differences in En morphological analysis by scanning electron microscopy (SEM) and by fluorescence microscopy (FM). The LUS samples analysis showed an important reduction in En in LUS of CS for the PDL group, in comparison with the elective CS group. The LUS overdistension, by fetal head malpositions (OPP, OTP, A) and malrotations, lead to dystocia, modification of vascularization, and En reduction. The En reduction in PDL suggests that drugs used during the LNA, usually local anesthetics and opioids, cannot control the “dystocic pain”, that differs from normal labor pain. The IU administration in labor and the consequent diagnosis of dystocia suggest stopping the numerous and ineffective top-up drug administration during LNA, and to shift the labor to operative vaginal delivery or CS.

## 1. Introduction

The uterus is an organ richly innervated by neuro fibers and neurotransmitters, mostly located in the cervix [1]. Neurofibers and neurotransmitters are detected mainly in the stroma and around arterial vessel walls in the myometrial and endometrial layers. They play important roles in physiological reproduction. Neurofibers and neurotransmitters are released after adrenergic or cholinergic nerve fibers stimulation. This is called sympathetic/parasympathetic co-transmission. Moreover, the cholinergic fibers have vesicles containing endogenous opioids involved in eutocic pregnancy, such as endorphins and enkephalins [1].

Malvasi et al. [2,3] demonstrated the presence of neuro fibers and neurotransmitters in the Lower Uterine Segment (LUS), in the uterine scar in greater density. Moreover, Malvasi et al. [4] showed that in Prolonged Dystocic Labor (PDL) there is a reduction in adrenergic and noradrenergic neuro fibers and neurotransmitters in the LUS.

These nerve fiber alterations observed above, generally, are identifiable in all PDLs and are the consequences of the LUS overdistension, such as changes in vascularization, ischemia, inflammation, cell apoptosis, and degeneration of neuro fibers and neurotransmitters [5].

The LUS suffers an important anatomical change in the second dystocic labor stage, especially when fetal malpositions and malrotations, such as Occiput Posterior Position (OPP), Persistent Occiput Posterior Position (POPP), Asynclitism (A), or Occiput Transverse Position (OTP), occur [6,7].

In the last thirty years, intrapartum ultrasonography (IU) proved to be a good diagnostic tool, associated with the vaginal digital examination, for fetal malpositions and malrotations detection in the birth canal [8].

Starting from these data, the aim of this experimental and morphological investigation is to quantify the Enkephalinergic (En) neuro fibers and neurotransmitters in the LUS of patients with PDL during Labor Neuraxial Analgesia (LNA) submitted to cesarean section (CS) for PDL, versus patients scheduled for elective CS.

## 2. Results

Of the 132 women enrolled, only 75 completed the study. The remaining 57 pregnant women were lost during the investigation for various reasons: 26 refused to undergo LUS biopsy, 13 women underwent urgent cesarean section at night (there was no histology laboratory available), and 18 patients subsequently refused to continue the study.

The pregnant women were homogenously subdivided into two groups as follows: 37 patients in group 1, as pregnant women scheduled for elective CS, and 38 women in group 2, as submitted to CS for PDL.

The mean age of the patients was similar in both groups, so no statistical differences were found for the mean BMI, gestational age of delivery, and birthweight for both groups (Table 1).

The examined histological samples to quantify the distribution of En nerve fibers demonstrated a significant and notable reduction in the concentration of En fibers in the LUS of patients with PDL (9 ± 2.7) in comparison with patients scheduled for elective CS (15 ± 1.4), where the LUS had not been subjected to stress for hours. All these are reported in Table 2 and in Figure 1.

## 3. Discussion

The endogen opioid system is a complex system that involves autonomous the Peripheral Nervous System (PNS) and the Central Nervous System (CNS), and somatic [9], which rules over the nociceptive system [10,11].

The endogenous opioid system includes many neurons localized in several parts of the human body, producing three different opioids: B-endorphin, met- and leu-enkephalins, and dynorphins [12].

These opioids are involved in the mechanism of pain control; indeed, they act as neuromodulators and neurotransmitters on three classes of receptors: delta, kappa, and mu [13]. Kojic et al., described the physiological bases and regulatory mechanisms involved in labor pain, including the role of opioids and their receptors [14].

Larsson et al., were among the first to isolate Met- and Leu-enkephalin from brain tissue, reporting separate populations of Leu- and Met-enkephalin nerves [15].

Subsequently, in animal models, Miller and Pickel confirmed the presence of enkephalinergic neuro fibers in the central nervous system and in the peripheral nervous system, by immunoreactive enkephalin staining [16].

Di Tommaso et al., demonstrated a differential localization of a large number of neurotransmitters in the non-pregnant uterine fundus, corpus, and, particularly, in the cervix, where the highest concentration of En neuro fibers was detected [17].

Some clinical and experimental studies have been conducted through the years demonstrating the presence of opioids in pregnancy. Since 1980, the presence of an En-like substance has been demonstrated in human cord blood and maternal, as well as in amniotic fluid [18].

Goebelsman et al., discovered that beta-endorphin levels increase during labor in general anesthesia but they do not change in epidural anesthesia [19].

Aurich et al., demonstrated, in an old study, the presence of better a-endorphin and met-En in cows’ plasma during gestation and at parturition, and in newborn calves [20]. Then, the results of a successive experimental study suggested that the reduced release of En in preterm calves could be related to a delayed adaptation to environment after delivery [21].

Belisle et al., described opioid receptors in the placenta [22], while Kornyei et al. [23] studied the effect of opioid neuropeptides on human myometrial cells in vitro. The authors demonstrated the inhibitory effect of opioid endogenous neuropeptides in the human uterus, in the regulation of cell growth and proliferation [24].

Hu et al. [25] demonstrated that fetal distress is associated with endogenous opioids, measuring concentrations of plasmatic endogenous opioid neuropeptides (EOP), such as beta-endorphin, endorphin A [1,2,3,4,5,6,7,8,9,10,11,12,13], and leu-enkephalin, measured by radioimmunoassay. They detected that the modifications in EOP levels in umbilical artery may play an important role in fetal distress, in the pathophysiological changes [25].

Malvasi et al., by intrapartum sonography, demonstrated that early epidural analgesia at a low dose does not determine dystocia, but the latter is due to cephalopelvic disproportion [26]. 

Similarly, Pizzicaroli et al., described that combined spinal–epidural analgesia determined a slowdown of the fetal head initial progression and rotation without influencing the type of delivery [27]. Moreover, the authors also reported a correlation, during operative vaginal delivery (OVD), between the angle of progression (AoP) degree and a positive outcome of vacuum application [27]. Carvalho et al., added that the AoP and head–perineum distance, determined by IU, correlate with the duration of labor, with a favorable correlation with the type of delivery prediction [28].

Malvasi et al. [29] demonstrated, furthermore, that AoP and Pubic arch angle in POPP and A are related to high OVD rates. In fact, during the second stage of labor, pregnant women diagnosed for POPP with A by IU, should stop the LNA administration, due to uselessness since the labor pain is related to the dystocia [29].

Indeed, Beck et al., reported that IU evaluation of fetal head progression, malposition, and rotation diagnosis during LNA can alert obstetricians about the possibility of a dystocic delivery, indicating stopping drugs during LNA and to switch to OVD or CS. This suggestion was to avoid fetal and maternal complications and possible medical-legal consequences, such as litigation, liability, and court claims [30].

Rowlands and Permezel disserted the many complex mechanism of pain during labor, that could be related to LUS distension and cervical dilatation [31]. The prolonged chronic LUS distension was inflammation due to mechanical stress leading to preterm labor. Furthermore, tissue remodeling and muscle growth were observed in response to this uterine overdistension [32].

Li et al., demonstrated that PGE2 and PGF2a are involved in the uterine inflammatory process during parturition, by stimulating the release of neutrophils, cytokine, and cyclooxygenase-2 products to intensify their own production [33].

Malvasi et al., suggested that a prolonged fetal head station in LUS, leading to overdistension and inflammation, is responsible for the denervation and reduction in neuropeptides in the dystocic LUS [3]. The same author also reported connective tissue modifications, in particular the reduction in laminin and collagen IV, in uterine samples of patients with PDL, compared with a uterine sample from patients with eutocic labor [34].

The central Importance of the cervix in labor pain is also suggested by the results of the Tingaker et al., investigation [35], in which a marked difference in the TRPV1 innervation between the corpus and the cervix was observed in the uterine sample of pregnant patients. In fact, while there was a spread of TRPV1 innervation through the whole uterus in non-pregnant patients, during labor, this innervation was concentered only in the cervix [35].

Furthermore, as above mentioned, the cervix is one of the regions of the uterus which harbors the vast majority and greatest variety of neuro fibers, especially in a pregnant uterus [17]. This is true not only for the enkephalinergic fibers, but also the oxytocin ones, highly present in the cervico-isthmic zone, influencing the reproductive system and sexual disorders after surgical cervical procedures [36]. On this important topic, Malvasi et al., investigated the presence of substance P (SP) and vasoactive intestinal polypeptide (VIP) and their fibers in the LUS of patients submitted to CS [2]. They showed that the levels of SP were higher in the LUS of patients scheduled for repeated CS, while the levels of VIP were reduced in the LUS. These results lead to hypothesize that the increase of SP in the post-CS LUS could be linked to cervical ripening, leading to dystocia during vaginal birth after CS (VBAC), while VIP reduction could determine the internal uterine orifice relaxation, worsening the LUS cervical ripening and formation [2].

Returning to the speech on LNA utilization during labor, as an important technique applied to obtain an efficient control of labor pain, Gupta and Partani examined these techniques in their review, discussing the advantages, drawbacks, and indications of each technique along with the intrathecal and epidural doses of drugs [37].

The problem continually arises in the labor room, when the patient in pain claims the use of the LNA and the obstetrician and midwife are skeptical about the benefits and disadvantages of labor progress and the influence that LNA may have on labor outcome.

Segal et al. [38] and Wang et al. [39] both showed that OVD incidence was lower in labor treated with low doses of anesthetics opioid mixture, while Wong et al. [40] described that the severe pain early onset and analgesic drugs high doses anticipate possible fetal distress and OVD.

From an obstetric point of view, in modern medical nomenclature, the second stage of labor is the time between complete cervical dilatation and delivery. American College Obstetrics and Gynecology defined the PDL for a nulliparous pregnant under LNA as more than three hours [41]. Nevertheless, a recent investigation suggested that obstetricians should increase the time limiting the vaginal delivery by one hour over the ACOG recommendation limit [42]. In addition, the International Society Ultrasound Obstetric Gynecology guidelines recommended the routine clinical use of IU in PDL [43], to describe, by IU images, a diagnosis of the fetal head malposition in OP, OPP, POPP, and Transverse A. These recommendations could be very useful in the case of the use of LNA during labor that is prolonged.

Several authors in experimental studies have used immunofluorescence to identify the En (enkephalinergic fibers) mainly in animal models and both in CNS [44,45,46,47] and PNS [48]; however, this is the first experimental study of human En (enkephalinergic fibers) of the uterus in PDL.

## 4. Materials and Methods

A total of 1156 nulliparous patients of the Department of Obstetrics and Gynecology of two University-affiliated Hospitals were evaluated from March 2012 to March 2016.

The inclusion criteria for pregnant women at initial labor were: patients with a single pregnancy at term, fetus in cephalic presentation, and no complications in pregnancy.

The exclusion enrolment criteria were pregnant women with any previous gynecologic surgery and any of the following problems experienced during pregnancy: macrosomia, infections, anticoagulation therapy, pre-eclampsia, HELLP syndrome, ruptured membranes for more than 36 h, placenta previa, and other placental pathologies.

A total of 158 women were assessed and declared eligible to participate in the study. Of these, 132 women provided informed consent and were enrolled, after an ultrasound evaluation on initial labor (Figure 2). All enrolled patients signed informed consent before inclusion in this study, as approved by the local institutional research ethics committee.

These pregnant women were submitted to intrapartum transabdominal sonography (ITAS) assessment for detection of head position in the first and second stage of labor, to detect either the POPP and the translation from anterior to posterior fetal head position. The fetal ultrasound (US) landmark to detect the OPP was the single fetal orbit, with “face to pubes” opposed on the presented fetal head side.

All pregnant women with PDL were patients in spontaneous labor and in LNA. The LNA was administered with a low dose of combined spinal-epidural (CSE) analgesia.

The CSE analgesia technique consisted of the needle-through needle technique in intervertebral space (L3-L4) (Espocan^®^-, B. Braun, Italia). The spinal needle “Spinocan^®^” 27 G was used through the Tuohy needle of 18 G. After that cerebrospinal fluid exited from the needle, and a mixture of ropivacaine 0.02% with 0.25 µg·mL^−1^ of Sufentanil (5 mL) was administered in the spinal space. The mixture of ropivacaine 0.07–0.15% (dilution of drugs depends on the stage, position, and station of the head) with 0.3 µg·mL^−1^ of Sufentanil (10–15 mL) was administered in epidural space. Top-ups were repeated every two hours and when the pain started again until full dilatation was reached.

A total of 132 patients were evaluated during the first and second stages of labor. They were in spontaneous labor and under CSE analgesia. These patients were followed all without active labor management by oxytocin use and rupture of membranes. Starting from 3 to 4 cm of cervical dilatation and fetal head at ischial spine station −1 or lower, women were assessed by ultrasonography (Aloka instrument SSD 2000 MultiView, Tokyo, Japan, GE Healthcare instrument, Voluson 730 Expert, Chalfont St. Giles, United Kingdom), both equipped with a multifrequency convex transabdominal transducer. All women were also examined by digital examinations (DE), at intervals of 45–90 min in the first, and of 15–30 min in the second stage. All women were followed until delivery, and all data were collected and recorded in a database, successively analyzed by an independent reviewer.

According to American College Obstetrics and Gynecology guidelines, labor as was defined as prolonged labor when it lasted more than 4 h [49].

The PDL was recorded when the duration of the second stage exceeded 4 h, according to International Guidelines, and showed an angle of progression ≤97°. The PL for the OPP, the asynclitism, or the OPT was diagnosed through vaginal examination (VE) and IU. The OPP was diagnosed by the face-to-pubis sign and orbit-to-pubis sign [50].

The asynclitism was diagnosed by the squint sign and asymmetric midline sign [51,52].

All patients demonstrating a PDL underwent a CS according to traditional obstetric criteria.

The control group was composed of women set to undergo non-urgent cesarean section for various reasons, under LNA, by CSE. These patients of the control group, undergoing non-urgent cesarean section, were pregnant women with a singleton fetus in a cephalic presentation to undergo elective CS, for previous at-term CS, scheduled after an ultrasonographic diagnosis of breech or transverse presentation, patients with pregnancy after assisted reproductive technique, multiple pregnancies, and women wishing to have a CS for personal reasons.

All patients, before CS, received a prophylactic antibiotic administration of 2 gr of Cefazolin intravenously. The surgical technique was a modified Stark’ CS: the uterine incision was transversally on the LUS after bladder flap detachment. After fetus extraction, the placenta was delivered spontaneously and the uterus was exteriorized. The surgeons sampled four serial consecutive full-thickness sections of 5 mm depth (with the inclusion of the myometrial layer) on the LUS, with the scissors for morphological analysis. Samples included the full thickness of the cervical wall, taken from each LUS using sterile scissors: two samples of approximately 5 mm depth were obtained from the upper and two from the lower edge.

The specimens were immediately transferred, in a container filled with dry ice, to the laboratory, where samples were washed by immersion in cold Krebs–Ringer’s solution, and assayed with immunofluorescent techniques to detect En-positive nerve fibers.

### 4.1. Detection of En Neuro Fibers

Slices were cut with a cryostat to obtain thin sections of 40 μm. Five consecutive serial sections were placed on five separate slides and prepared for the detection of each neurotransmitter. On the first slide, the pathologist detected the LUS sample. On the second slide, excess primary or secondary antiserum was omitted, denatured, or previously absorbed by the corresponding peptide. On the third slide, primary or secondary antiserum was replaced by a nonimmune serum. On the fourth slide, the sample was fixed by immersion in a 4% formaldehyde solution with phosphate buffered saline (PBS), which did not preserve the immunoreactive sites. On the fifth slide, the samples were denatured with formaldehyde before or after treatment with the primary antiserum or prior to treatment with the secondary antiserum. Polyclonal antibodies for the En immunohistochemical assays were prepared according to the instructions of the Phoenix Pharmaceuticals catalogue (n° H-024-21 and n° H-051-01, respectively).

Fluorescence staining was performed by the following method: En was detected using rabbit anti-EN serum (Phoenix Pharmaceuticals, Inc., Burlingame, CA, USA) in lyophilized form reconstituted in 50 μL of diluted water (the equivalent of undiluted antiserum) without refreezing any portions. Once reconstituted, the antibody was stable for a few days at −4 °C. Tissue samples were incubated in auto-fluorescent antibody (diluted to 1:100 in PBS) for 18–24 h at room temperature and examined via fluorescence microscopy using Leitz Ortoplan microscope (Leica Microsystems, Wetzlar, Germany) equipped with an epi-illumination system. The light source was from a mercury lamp (HB, 100) combined with selective Leitz filters.

Stained samples were washed in PBS, immersed in Entellan mounting medium (non-auto fluorescent), and examined using a Zeiss III photomicroscope (Carl Zeiss AG, Oberkochen, Germany). Once the nerve fibers were marked for En, it was possible to identify the area of each fiber marked by a specific fluorescent neurotransmitter via light microscopy. Morphometrics is a validated method for quantification of the density of cellular elements indicative of different pathological and physiological conditions in the same tissue.

### 4.2. Data Analysis on Neuro Fibers

The density of En fibers was calculated by quantitative analysis using the Quantimet Leica 2000 image analyzer with the following parameters:Number of En-positive fibers counted in 10 randomly chosen fields;Percentage of the total area occupied by those fields;Number of observed varicoses;Number of crossings or intersections of the nerve fibers;The total perimeter of En-positive structures in proportion to an average value (100 for each field).

Morphometrical quantification of the density of each type of nerve fiber was performed on photographs of stained samples using a Quantimet Leica 2000 image analyzer (Quantimet 500 Leica Microsystems Imaging Solutions Ltd., Cambridge, UK). The software instrument can count and express these fluorescent areas in conventional units (C.U.).

This parameter indicates the percentage of the area occupied by a single type of nerve fiber about the total observed area. By adding these values together (a single type of nerve fiber), it is possible to evaluate (in C.U.) the sum of the areas occupied by various types of neuro fibers.

The software also calculates the average values and reduces them to just one value with standard deviation: this value can be read on the instrument display and is reported with the standard error of the mean (SEM).

### 4.3. Statistical Evaluation

Statistical analysis was performed on the data obtained from the measurements of each sample by the Quantimet Leica 2000 analyzer. The data were averaged to obtain a median value per case and the mean ± SD were then calculated for each nerve fiber group. Repeated immunofluorescent controls were analyzed using one-way analysis of variance with Bonferroni correction for multiple comparisons of normally distributed data. Statistical calculations were performed with MedCalc software (version 11.4.1.0; http://www.medcalc.org/). Statistical significance between immune-histochemical samples groups was determined using Student’s *t*-test. A *p* value of <0.05 was considered statistically significant.

## 5. Conclusions

When the obstetric course of labor is not favorable and it turns into a prolonged labor, as indicated by the ACOG guidelines, it should be recommended to stop the LNA, since the PDL determines an LUS overdistension, ischemia, inflammation, and neuro fibers reduction, in particular, of the enkephalinergic neuro fibers, reported in Figure 1. The indication to suspend LNA in PDL patients is not due to the negative effect of LNA on labor, but to the need not to prolong labor beyond 4 h, to reduce maternal and fetal complications. The PDL evokes a particular kind of pain, a “dystocic pain”, which does not respond to analgesic and opioid drugs, commonly used during a CSE analgesia. It happens because Leu- and Met-enkephalin neuro fibers are depleted for the mechanical stress exercised to the LUS by the fetal head prolonged position in the PDL. The En reduction was shown in Figure 3.

In this image, the En detection in SEM images from LUS samples of patients with PDL is significantly reduced when compared to the LUS samples of patients scheduled for elective CS. Therefore, it should be recommended to promote the type of delivery without an LNA prolonging, as it should be maternal to maternal-fetal complications.

## Figures and Tables

**Figure 1 ijms-24-03767-f001:**
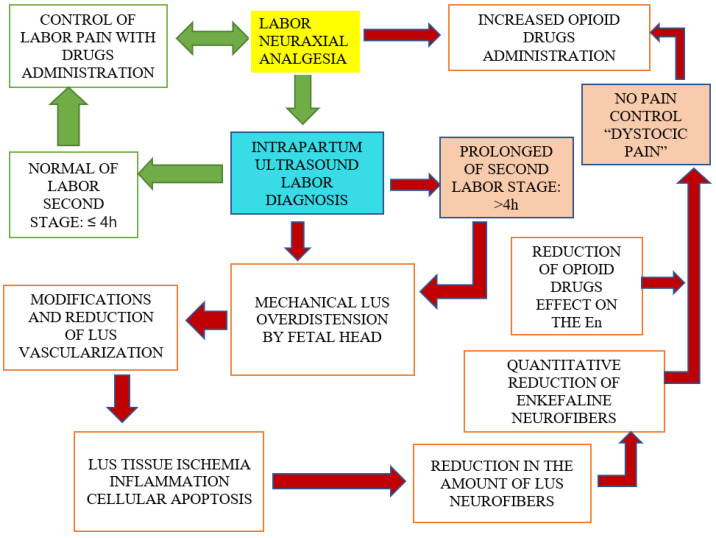
Proposed mechanism of Enkephalin neuro fibers effect of normal and dystocic prolonged labor (DPL) with labor neuraxial analgesia (LNA).

**Figure 2 ijms-24-03767-f002:**
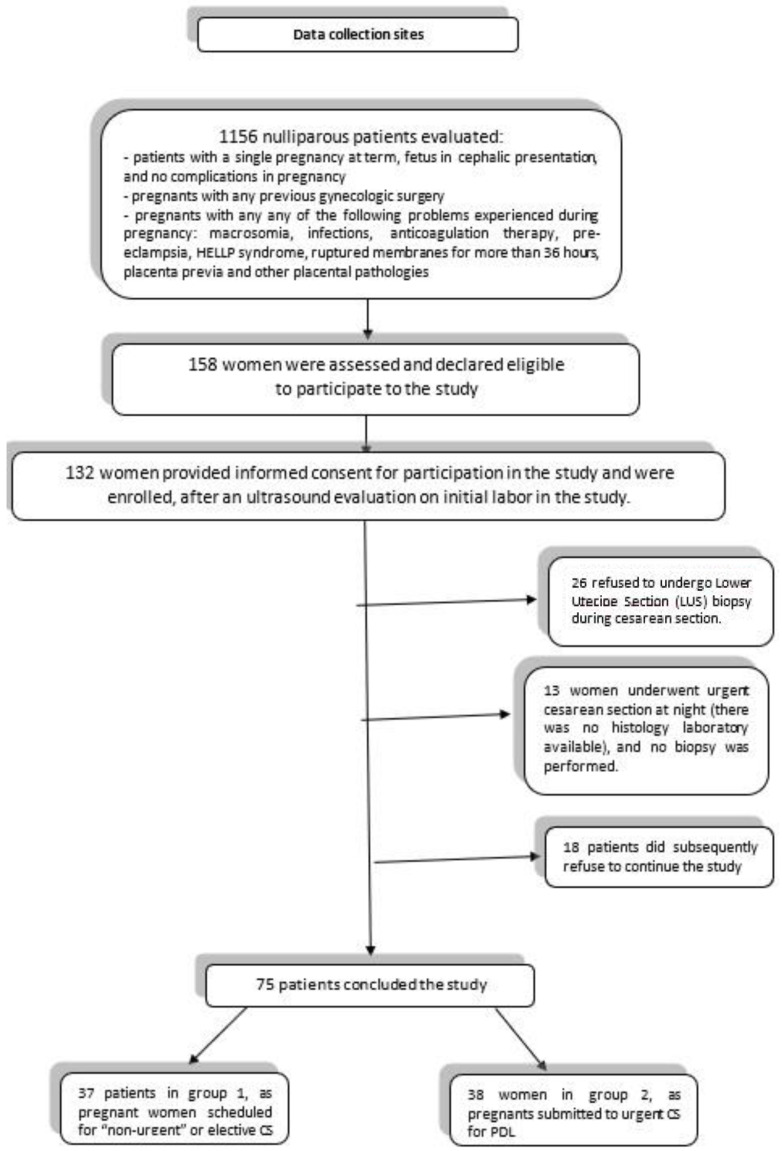
The flow chart of the study.

**Figure 3 ijms-24-03767-f003:**
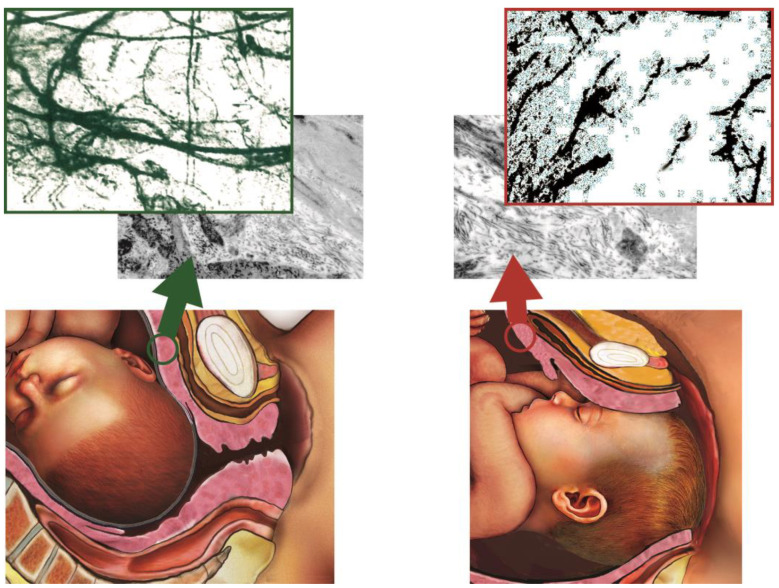
The images show a sectional drawing of the lower uterine segment (LUS), the left one of the elective CS (ECS), and the right one of the dystocic prolonged labor (PDL). At the top of the figure, there are the corresponding microscopic photos of LUS in ECS (top left) and in PDL (top right) taken in scanning electron microscopy (SEM) with the collagen fibers. Over the SEM photos, there are the imagines of the enkephalinergic fibers of the same LUS section respectively on the left in the ECS and on the right in the PDL, obtained with immunofluorescence. You can see that the En fibers visualized in the specimens of LUS in DPL are less the those in ECS.

**Table 1 ijms-24-03767-t001:** Demographic characteristics of two groups.

	GROUP 1	GROUP 2	*p* Value
	Elective Cesarean Section (n = 37)	Cesarean Section in Prolonged Dystocic Labor (n = 38)	
Age (year)	34.7 ± 5.3	36.8 ± 2.7	>0.05
BMI (Kg/m^2^)	27.9 ± 2.6	28.7 ± 2.5	>0.05
Gestational week (week + days)	39 ± 6.1	40 ± 7.8	>0.05
Birthweight (g)	3350 ± 329	3447 ± 302	>0.05

**Table 2 ijms-24-03767-t002:** Evaluation of Enkephalin (En) nerve fibers in the LUS of elective CS and in LUS of patients in PDLs.

GROUP 1	GROUP 2	*p* Value
Elective CS (n = 37)	Prolonged dystocic labor CS (n = 38)	
Nerve fibers density containing enkephalinergic (En) immune reactivity within specimens of human LUS		
LUS specimens 15 ± 1.4 (immune reactivity)	LUS specimens 9 ± 2.7 (immune reactivity)	<0.05

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
