# Peer review of "Prolonged Dystocic Labor in Neuraxial Analgesia and the Role of Enkephalin Neurotransmitters: An Experimental Study"

_ijms, 2023, doi:10.3390/ijms24043767_

Round 1

Reviewer 1 Report

Dear authors, I've read your manuscript and found it very interesting however I have some questions and remarks about your manuscript:

1.       Manuscript structure: First of all, “materials and methods” has to be after introduction but not after results and discussion.

2.       If the study approved by ethics committee - No. of approval has to be presented in the manuscript.

3.       The study period is 2006-2010, that means more than 12years old. During this study period 1156 nulliparous patients were evaluated, but only 158 were assessed to participate and 75 completed the study (very low sample size)? May be presented flow chart could be more clear to the readers?

4.        Grouping of the patients is also confusing as in the text:

a.       71-73 rows –The pregnant woman has been homogenously subdivided into 2 groups as follows: 37 patients in group 1, as pregnant women scheduled for elective CS, and 38 women in group 2, as submitted to CS for PDL.

b.      227-229 rows - All pregnant women were subdivided into two groups: patients in spontaneous labor and in LNA.

c.       238-240 rows - One hundred thirty-two patients were evaluated during the first and second stages of labor and were allocated into two groups: group I in spontaneous labor, and group II in labor by CSE analgesia.

d.      259-262 rows - The control group was composed of patients with a singleton fetus in a cephalic presentation to undergo elective CS, for previous at-term CS, scheduled for diagnosis of breech or transverse presentation, patients with pregnancy after assisted reproductive technique, multiple pregnancies, women wishing to have a CS for personal reasons.

5.       It seems, that 1group patients (elective CS) were not in labor at all. Also there is no protocol of anesthesia method for them. This comparison of study groups raises more questions than gives the answers.

6.       Table 2 presents results of LUS specimens which has no statistical difference P > 0,05  rows 84-86. Also there are values with no dimensions in this table.

7.       Considering all mentioned above, conclusions of the study are not associated with you study results and subsequently not convincing the reader.

8.       Row 50; 55, 63 – “uterine” word unnecessary as LUS abbreviation is - low uterine segment.

9.       Row 134 there is unnecessary indefinite article “a”

10.   Row 167 – “While patients”?

11.   More than half -27 references are older than 10 years. 

Author Response

  1. Manuscript structure: First of all, “materials and methods” has to be after introduction but not after results and discussion.

Answer: The Journal requires this setup in the manuscript setup. We agree with the reviewer, but if the reviewer looks at the format of the Journal, this is requested by the editorial office. Therefore, this comment, appropriate in our opinion, must be transferred to the editorial office (which we must comply with for the formatting rules of the manuscript).

  1. If the study approved by ethics committee - No. of approval has to be presented in the manuscript.

Answer: We specified in the final part of the manuscript that the study was approved by the ethics committee in the section Institutional Review Board Statement.

  1. The study period is 2006-2010, that means more than 12 years old. During this study period 1156 nulliparous patients were evaluated, but only 158 were assessed to participate and 75 completed the study (very low sample size)? May be presented flow chart could be clearer to the readers?

Answer: The reviewer is absolutely right, but we need to explain why these numbers. We apologize for the typographical error, but the dates shown in the text are incorrect. The study was performed from 2012 to 2016, the study funding period. Then, immunohistochemical studies with monoclonal antibodies are expensive and it is possible to examine a limited sample of patients. For which the disbursement of funds was divided into small parts. We were then forced to select a limited number of patients due to the very high costs of monoclonal antibody kits. This was specified in the manuscript text, et the Results section: Of the 132 women enrolled, only 75 completed the study. The remaining 57 pregnant women were lost during the investigation for various reasons: 26 refused to undergo LUS biopsy, 13 women underwent urgent cesarean section at night (there was no histology laboratory available), and 18 patients did subsequently refuse to continue the study. The flow chart was included in the manuscript revision.

  1. Grouping of the patients is also confusing as in the text:

71-73 rows –The pregnant woman has been homogenously subdivided into 2 groups as follows: 37 patients in group 1, as pregnant women scheduled for elective CS, and 38 women in group 2, as submitted to CS for PDL.

Answer: We omitted to specify, again after the abstract, the acronym PDL as prolonged dystocic labor.

227-229 rows - All pregnant women were subdivided into two groups: patients in spontaneous labor and in LNA.

Answer: We probably weren't clear in exposing the text that the patients with PDL were all in Labor Neuraxial Analgesia (LNA). We have clarified in the text

  1. 238-240 rows - One hundred thirty-two patients were evaluated during the first and second stages of labor and were allocated into two groups: group I in spontaneous labor, and group II in labor by CSE analgesia.

Answer: We agree with the reviewer. Again, we were not clear. We have corrected the text. The PDL patients were all in CSE. One hundred thirty-two patients were evaluated during the first and second stages of labor, they were all in spontaneous labor and in labor by CSE analgesia.

259-262 rows - The control group was composed of patients with a singleton fetus in a cephalic presentation to undergo elective CS, for previous at-term CS, scheduled for diagnosis of breech or transverse presentation, patients with pregnancy after assisted reproductive technique, multiple pregnancies, women wishing to have a CS for personal reasons.

Answer: We clarified in the text that all patients in the control group were all women undergoing non-urgent cesarean section vs urgent cesarean section for PDL.

  1. It seems, that 1 group patients (elective CS) were not in labor at all. Also, there is no protocol of anesthesia method for them. This comparison of study groups raises more questions than gives the answers.

Answer: we were not clear in the exposition, specifying that the patients of the control group were women to undergo cesarean section (for various reasons), who arrived at the hospital and were subjected to non-urgent cesarean section in LNA by CSE. We clarified in the text.

  1. Table 2 presents results of LUS specimens which has no statistical difference P > 0,05 rows 84-86. Also, there are values with no dimensions in this table.

Answer: Statistical significance between immune-histochemical samples groups was determined using Student's t-test. It was reported into the manuscript revision.

  1. Considering all mentioned above, conclusions of the study are not associated with you study results and subsequently not convincing the reader.

Answer: We totally agree with the reviewer. In the manuscript there are unclear points, which we have modified, upon correct reporting by the reviewer. There were terminology errors. In both groups studied, the patients used the LNA as an analgesia technique: group 1, made up of women undergoing "non-urgent" cesarean section and group 2, made up of nulliparous women with PDL (labor lasting more than 4 hours, as per ACOG guidelines).

  1. Row 50; 55, 63 – “uterine” word unnecessary as LUS abbreviation is - low uterine segment.

Answer: According to reviewer comment, it was deleted.

  1. Row 134 there is unnecessary indefinite article “a”

Answer: ok According to reviewer comment, it was deleted.

  1. Row 167 – “While patients”?

Answer: According to reviewer comment, it was deleted.

  1. More than half -27 references are older than 10 years. 

Answer: We agree with the reviewer, but there is a basic problem with the current literature. A literature review was performed with the keywords: enkephalins, neurotransmitters, prolonged labor, dystocic labor, delivery-analgesia. Unfortunately, the current literature is scarce of current works. That’s the reason why the references are older than 10 years.

Reviewer 2 Report

This study demonstrating histomorphological change of enkephalinergic neuro fibers in uterine of patients under prolonged dystocic labor is interesting. However, reduction of susceptibility against opioid is not clear. Chronological changes in pain score and amount of rescue analgesic drugs during prolonged labor in comparison with normal labor under same combined spinal-epidural analgesia should be shown. 

Author Response

This study demonstrating histomorphological change of enkephalinergic neuro fibers in uterine of patients under prolonged dystocic labor is interesting.

However, reduction of susceptibility against opioid is not clear.

Chronological changes in pain score and amount of rescue analgesic drugs during prolonged labor in comparison with normal labor under same combined spinal-epidural analgesia should be shown. 

Answer: We agree with the reviewer, so let's try to clarify what we have studied. The study demonstrates a reduction of enkephalins in prolonged labor (PDL) vs patients in normal labor (without prolongation of labor hours), which is the experimental morphometric data we investigated. The authors of the research hypothesized a lower presence of opioids in the LUS and, consequently, a lower efficacy of opioid drugs on the LUS in prolonged labor (PDL). Therefore, in the nullipara with prolonged labor (PDL), due to malposition of the fetal head (documented with intrapartum ultrasound) it is completely useless to continue with the administration of analgesics after 4 hours (prolonged labor, as indicated by the ACOG guidelines). We did not verify the chronological changes in pain score and the amount of rescue analgesic drugs in PDL, in comparison with normal labor, under same CSE analgesia. This is an excellent starting point for a future study, following this one.

Reviewer 3 Report

The authors studied innervation of the lower uterine segment and demonstrated difference in innervation between those women who develop dystocia and those who do not.  Given the time period in which the muscle biopsy was performed, it is not possible to determine if the altered innervation resulted in the dystocia or did the dystocia cause the altered innervation.  The conclusion about anesthesia is well beyond the results; there is a difference in innervation and this difference does not mean that top-up and neuraxial analgesia should be abandoned.

Abstract:  Change "The investigation detected" to "This investigation studied"

The discussion needs to be more focused - what are the specific points you are trying to make?

The materials and methods should follow the introduction.

Your results do not allow you to conclude about analgesia and its management.  This conclusion is well beyond the results.  

Author Response

The authors studied innervation of the lower uterine segment and demonstrated difference in innervation between those women who develop dystocia and those who do not. Given the time period in which the muscle biopsy was performed, it is not possible to determine if the altered innervation resulted in the dystocia or did the dystocia cause the altered innervation. The conclusion about anesthesia is well beyond the results; there is a difference in innervation and this difference does not mean that top-up and neuraxial analgesia should be abandoned.

Answer: We try to clarify what was reported in the study. During the second stage of labor, the women with prolonged labor dystocia (PDL) underwent intrapartum ultrasound (IU) every 30 minutes and malposition and malrotation of the fetal head (occipito-posterior, transverse, asynclitic), which resulted in dystocia with overdistention of the LUS. Therefore, from what we have evaluated in the study, dystocia is the cause of the impaired innervation of the enkephalinergic fibers. Furthermore, the ACOG guidelines establish that after 4 hours of labor, in the second stage of labor, delivery must be completed to avoid maternal and fetal complications. Therefore, in these women in prolonged labor it is useless to continue LNA by means of CSE, because pain control does not resolve the malposition of the fetal head in the birth canal and, therefore, the doctor must decide how to carry out the operative delivery, with suction cup, with forceps or by caesarean section. At this point, the anesthetist, in agreement with the obstetrician, will be able to carry out the conversion from analgesia to anesthesia, always in CSE.

Abstract:  Change "The investigation detected" to "This investigation studied"

Answer: According to reviewer comment, it was modified.

The discussion needs to be more focused - what are the specific points you are trying to make?

Answer: We try to clarify what is reported in the study. In patients with prolonged labor (PDL) under LNA >4h (according to ACOG guidelines) we observed a reduction of enkephalinergic fibers due to overdistension of the LUS due to malposition or rotation of the fetal head (after ultrasound monitoring of the progress of labor). The indication to suspend LNA in PDL patients is not due to the negative effect of LNA on labor, but to the need not to prolong labor beyond 4 hours, to reduce maternal and fetal complications. This was reflected in the manuscript edits, in the conclusions

The materials and methods should follow the introduction.

Answer: The Journal requires this setup in the manuscript setup. We agree with the reviewer, but if the reviewer looks at the format of the Journal, this is requested by the editorial office. Therefore, this comment, appropriate in our opinion, must be transferred to the editorial office (which we must comply with for the formatting rules of the manuscript).

Your results do not allow you to conclude about analgesia and its management. This conclusion is well beyond the results. 

Answer: We shared the reviewer's concerns and what was indicated was reported in the changes to the manuscript, in the conclusions.

Round 2

Reviewer 1 Report

Thanks for authors for their efforts to improve the manuscript. It looks really better now. However: 81-84rows „The examined histological samples to quantify the distribution of En nerve fibers demonstrated a significant and notable reduction in the concentration of En fibers in the LUS of patients with PDL (9 ± 2.7) in comparison with patients scheduled for elective CS 83 (15 ± 1.4), where the LUS had not been subjected to stress for hours”

Significant reduction seems to be then statistical significance is evident p < 0.05.  Remains unclear is it so in your study as in table 2: p > 0.05? 

Author Response

Answer: we apologize to the reviewer, but we realized that there was a typographical error in the table. Certainly, the p is <0.05 and not > 0.05. Unfortunately, in the revision of the text we have now noticed the error, which we immediately corrected, as rightly indicated.

Reviewer 2 Report

The present study did not demonstrate that the reduction of enkephalinergic neuro fibers in lower uterine segment induces the ineffectiveness of analgesic effects of opioid to prolonged labor pain. This point should be described as a limitation of the study.

Author Response

Answer: the aim of the study, as we reported in the introduction, was not to demonstrate the ineffectiveness of opioid drugs on the LUS, but the distribution of enkephalinergic fibers in the LUS of patients with PDL during Labor Neuraxial Analgesia (LNA) submitted to cesarean section (CS) for PDL, versus patients scheduled for elective CS (page 2, lines 63-65).

However, it can be speculated that the reduction of these fibers in prolonged dystocic labor, due to an overdistension of the LUS, with subsequent hypoxia and ischemia, may have a detrimental effect on the drugs administered in the LNA. That is, even if analgesics are administered with LNA in patients with LNA, as there are fewer neurofibers in the LUS subjected to prolonged stress (significant reduction of neurofibers), the efficacy on pain is almost limited by the reduction of neurofibers. I do not believe that what is reported is a limitation of the study, as the aim was not that indicated by the reviewer, but rather a biologically reasoned speculation based on other studies already published.